# Metabolic Difference Analysis of *Clostridium cellulovorans* Grown on Glucose and Cellulose

**Wen-Zhu Tang** [1,†], **Dan-Dan Jiang** [1,†], **Yi-Xuan Fan** [2] , **Quan Zhang** [3], **Li-Cheng Liu** [2], **Fu-Li Li** [2] and **Zi-Yong Liu** [2,*]

1   School of Biological Engineering, Dalian Polytechnic University, Dalian 116034, China
2   Shandong Provincial Key Laboratory of Synthetic Biology, Key Laboratory of Biofuels, Qingdao Institute of Bioenergy and Bioprocess Technology, Chinese Academy of Sciences, Qingdao 266101, China
3   Sinopec Dalian Research Institute of Petroleum and Petrochemicals, Dalian 116045, China
*   Correspondence: liuzy@qibebt.ac.cn; Tel.: +86-532-8066-2656; Fax: +86-532-8066-2778
†   These authors contributed equally to this work.

**Abstract:** As an anaerobic butyrate-producing bacterium, *Clostridium cellulovorans* can secrete a variety of extracellular enzymes to degrade plant-based cellulose. However, with glucose as the carbon source, it still secretes a large amount of protein in the broth. The metabolism and regulation are obscure and need to be further studied. Hence, in this study, *C. cellulovorans* was used to conduct fed-batch fermentation of glucose and microcrystalline at pH 7.0 to produce a higher level of butyrate in the bioreactor. It produced 16.8 mM lactate, 22.3 mM acetate, and 132.7 mM butyrate in 72 h during glucose fermentation. In contrast, it produced only 11.5 mM acetate and 93.9 mM butyrate and took 192 h to complete the fermentation with cellulose as the carbon source. Furthermore, there was no lactate detected in the broth. The analysis of carbon source balance and redox balance showed that 57% of the glucose was consumed to form acids in glucose fermentation, while only 47% of the cellulose was used for acid generation in the cellulose fermentation. Meanwhile, a large amount of protein was detected in the fermentation broth in both glucose ($0.9 \pm 0.1$ g/L) and cellulose ($1.1 \pm 0.2$ g/L) fermentation. These results showed that protein was also a main product. *C. cellulovorans* metabolized glucose to generate intermediate metabolites and reducing powers (NADH and $Fd_{red}$), then protein and acid synthesis consumed this reducing power to maintain the carbon source balance and redox balance in the cell metabolism. The results of comparative transcriptomics and comparative proteomics also supported the above conclusion. The method of studying the protein during *Clostridium* species fermentation provides a new perspective for further study on metabolic regulation.

**Keywords:** butyrate; extracellular protein; carbon source balance; redox balance; metabolic regulation





## 1. Introduction

Lignocellulose is the most abundant sustainable energy and can provide around 10% of global primary energy for social activities [1,2]. As an anaerobic cellulolytic bacterium, *Clostridium cellulovorans* is able to produce acids (including formate, acetate, lactate, and butyrate) and gases (including $CO_2$ and $H_2$) by degrading complex carbohydrates such as cellulose, xylan, and pectin, which are main components of lignocellulose [2–4]. It has also been engineered to be a butanol-producing strain by integrating an exogenous aldehyde/alcohol dehydrogenase [5–7]. Thus, *C. cellulovorans* has become a promising candidate for producing industrial biofuels and biochemicals with the utilization of ligno-cellulosic biomass, which is of great significance in cutting the cords to fossil energy [8].

*C. cellulovorans* can adjust the metabolic pathways and product synthesis in response to changes in the fermentation environment [9,10]. Butyrate is the main product in the fermentation process of *C. cellulovorans,* and its metabolic pathway has been outlined clearly as shown in Figure 1 [2,11,12]. There are four biochemical reactions that occur

during the conversion from acetyl-CoA to butyryl-CoA, including two redox reactions during which NADH acts as the sole electron donor. Then, butyrate is synthesized by phosphotransbutyrylase and butyrate kinase accompanied by ATP generation. Acetate and butyrate are the main products at pH 6.5, while the products include formic acid, lactate, and ethanol in addition to acetate and butyrate without pH control. Nevertheless, how pH affects the regulation mechanism of *C. cellulovorans* is still unclear. For the byproducts, the synthesis of acetate and formate does not require NADH, while lactate and ethanol require different amounts of NADH compared to that of butyrate synthesis (Figure 1). These phenotypes indicate that *C. cellulovorans* redistributes carbon sources and NADH to maintain the carbon balance and redox balance according to the pH change in the cell metabolism. Relative to the diversity of products, there are not many forms of reducing equivalents (mainly NADH and reduced ferredoxin) in the fermentation of anaerobic *Clostridium* species. As another reducing equivalent, reduced ferredoxin ($Fd_{red}$) is mainly reoxidized by $H_2$ production, which suggests that the $H_2$ concentration should also be determined as an important product in the redox balance analysis [5,12]. Significantly, electrons can be transferred between $Fd_{red}/Fd_{ox}$ and $NADH/NAD^+$ via a bifurcation reaction (Bcd/Etf complex, Figure 1), which is a unique characteristic of butanol/butyrate-producing clostridia. Thus, it is an effective strategy to study the metabolic regulation mechanism of *Clostridium* species by analyzing the production and distribution mechanism of reducing equivalents [13,14].

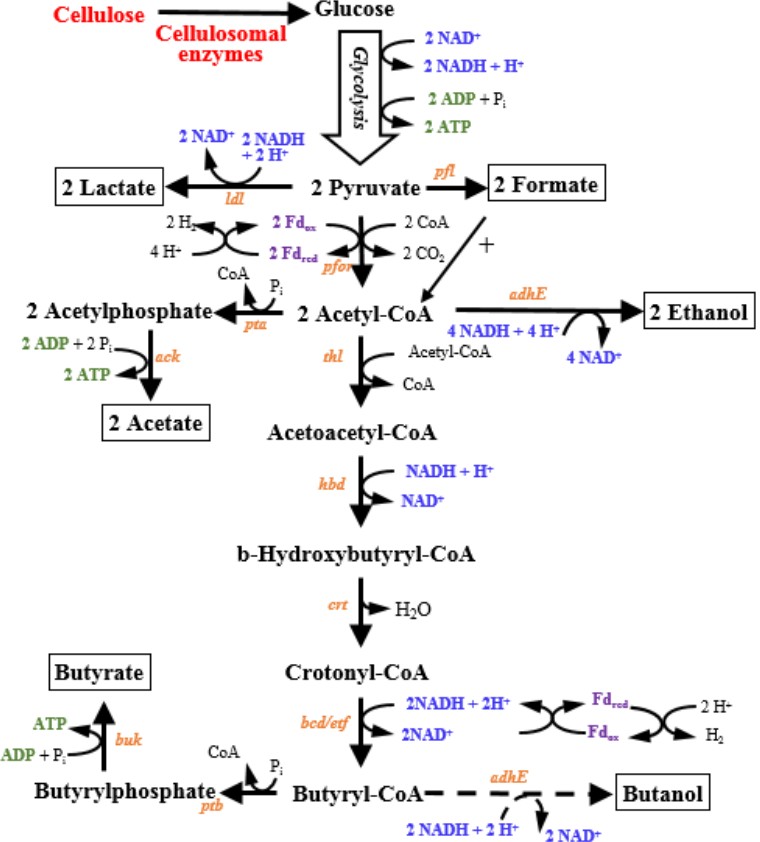

**Figure 1.** Metabolic pathway of acid biosynthesis in *C. cellulovorans*. Abbreviations: *pta*, phosphate acetyltransferase; *ack*, acetate kinase; *ptb*, phosphate butyryltransferase; *buk*, butyrate kinase; *adhE*, aldehyde/alcohol dehydrogenase; *pfor*, pyruvate:ferredoxin oxidoreductase; *ldh*, lactate dehydrogenase; *pfl*, pyruvate formate lyase; *thl*, thliolase/acetyl-CoA acetyltransferase; *hbd*, 3-hydroxybutyryl-CoA dehydrogenase; *bcd*, Acyl-CoA dehydrogenase; *etf*, electron transfer flavoprotein; *crt*, crotonase/short-chain-enoyl-CoA hydratase; $Fd_{red}$, reduced ferredoxin; $Fd_{ox}$, oxidized ferredoxin.

In order to obtain fermentable sugars from lignocellulose, *C. cellulovorans* secretes a quantity of carbohydrate activity enzymes (CAZys) to degrade its complex structure [15]. The CAZys have a variety of types, functions, and combinations that can act synergistically on the degradation. Some CAZys compose cellulosomes that are expressed at the cell surface. While there are also some enzymes called non-cellulosomal enzymes, these do not belong to cellulolytic enzymes but play a role in lignocellulose depolymerization [16]. These CAZys have been deeply studied with the completion of genome sequencing and application of omics technology in *C. cellulovorans* [2,17–20]. The genome data show that *C. cellulovorans* has a broader substrate spectrum and secretes 37% more types of CAZys than those of *Clostridium thermocellum,* which is recognized as the most efficient cellulose-degrading bacterium [21]. In addition, some research results also showed that the composition and production of CAZys varied widely with lignocellulosic biomass and soluble sugars. Theoretically, the synthesis of CAZys occupies the carbon source and energy during bacterium metabolism and will further affect the product production and redox balance. However, there is a gap in the experimental data on how CAZy synthesis affect the titer and yield of products. In the present study, we utilized two representative carbon sources, glucose and crystalline cellulose (Avicel in this study), in *C. cellulovorans* fermentation. Glucose can be metabolized directly through glycolysis, while Avicel must be degraded to soluble glucose by CAZys secreted by *C. cellulovorans* before it can be used as a carbon source. The relationship between products and CAZy synthesis was investigated by comparing the product concentration, carbon source and energy balance, functional genes, and protein expression under the two fermentation conditions. Because pH affects the product synthesis and growth, this study was carried out under a fixed pH condition at 7.0.

## 2. Materials and Methods

### 2.1. The Culture Medium and Batch Fermentation

*C. cellulovorans* 743B was purchased from the Deutsche Sammlung von Mikroorganismen und Zellkulturen GmbH (DSMZ), Braunschweig, Germany, and stored by freezing mid-exponential phase cultures with 30% glycerol at $-80$ °C. *C. cellulovorans* was revived and grown in a CM3 medium that contained (per liter): $(NH_4)_2SO_4$, 1.3 g; $KH_2PO_4$, 1.5 g; $K_2HPO_4 \cdot 3H_2O$, 2.9 g; $FeSO_4 \cdot 7H_2O$, 2 mg; $MgCl_2 \cdot 6H_2O$, 0.2 g; $CaCl_2 \cdot 2H_2O$, 75 mg; cysteine-HCl, 0.5 g; $NaHCO_3$, 0.5 g; SL-10 trace metal elements (1000×), 1 mL; yeast extract, 2 g; and cellobiose, 6 g. The composition of the fermentation medium was as follows (per liter): $(NH_4)_2SO_4$, 1.5 g; $K_2HPO_4 \cdot 3H_2O$, 1.0 g; KCl, 0.5 g; $MgSO_4 \cdot 7H_2O$, 0.5 g; $CH_3COONa$, 1.6 g; Cys-HCl, 2 g; yeast extract, 0.5 g; tryptone, 0.5 g; $MnCl_2 \cdot 4H_2O$, 25 mg; $CuCl_2 \cdot 2H_2O$, 2.5 mg; $CoCl_2 \cdot 2H_2O$, 25 mg; $ZnCl_2$, 10 mg; $H_3BO_3$, 2.5 mg; $NiCl_2 \cdot 6H_2O$, 2.5 mg; and $NaMoO_4 \cdot 2H_2O$, 2.5 mg. Glucose or cellulose (microcrystalline, Avicel PH-101, Sigma-Aldrich Inc., St. Louis, MO, USA, in this study) was added into the medium as the carbon source at an appropriate concentration. During the medium preparation, the operation process was kept anaerobic through nitrogen replacement and an anaerobic workstation (Coy Laboratory, Grass Lake, MI, USA).

The *C. cellulovorans* cells (5 mL) stored in glycerol bottles were inoculated into 50 mL of CM3 medium. After overnight cultivation, the culture was injected at 5% inoculum into a 5 L bioreactor (Infors Biotechnology Co., Ltd. Beijing, China) with a working volume of 2 L. Then, the fermentation process was kept at 36 °C and pH 7.0 with 2M KOH added automatically. Glucose (40 g/L) or Avicel (40 g/L) was used as a carbon source in the initial medium to study the effects of different carbon sources on fermentation product formation and metabolic regulation. An appropriate amount of glucose was added to the broth using a peristaltic pump to maintain the concentration at not less than 10 g/L, while Avicel (20 g) was added to the broth at 48 h and 72 h using a syringe to maintain an anaerobic environment. Three independent fermentations were performed for each growth condition of glucose- or Avicel-supplemented medium.



## 2.2. Analytical Methods

Cell growth was monitored via the absorbance at 600 nm ($A_{600}$) and determined using a Ultrospec 2100 Pro spectrophotometer (GE, Fairdield, CT, USA) during glucose fermentation at regular time intervals. The growth during cellulose fermentation was indicated by identifying the total amount of intracellular protein. The standard curve of the corresponding relationship between the absorbance ($A_{600}$) and intracellular protein concentration was established as follows: take out the cells in exponential phase during glucose fermentation and measure the OD value; collect the cells by centrifuging at 4 °C for 15 min, and wash the pellets twice with 0.9% ($w/v$) NaCl. Next, the cells were resuspended in different volumes to obtain different gradients of protein concentrations and treated using ultrasonication for 30 min. Then, 1 mL of the mixture was withdrawn and mixed with 1 mL of NaOH (0.1 M). The mixture was boiled for 10 min and centrifuged at $8000 \times g$ at 4 °C for 20 min. Then, the protein concentration in the supernatant was concentrated 10 times with a SpeedVac machine (CVE-2100 EYELA, Shenzhen, China) and determined by using a Bradford reagent (Sangon Biotec, Shanghai, China) following the manufacturer's instructions. Bovine serum albumin (BSA) was used as the standard. The standard curve of OD value and protein concentration was prepared according to measured data (Figure S1). Hence, the growth in cells during cellulose fermentation was measured by determining of intracellular protein concentration described above.

The total extracellular protein in the broth was determined by using the Bradford method described above. In addition, the total extracellular protein in the fermentation broth was determined by using the following weighing method. The broth was centrifuged for 20 min at 4 °C to remove precipitation, and the supernatant was transferred to a new beaker. The supernatant (1 L) was mixed with a 2-fold volume of ethanol for 12 h. The precipitated protein was obtained via centrifugation and transferred to a −80 °C freezer for 24 h. Then, the dry weight of the protein was determined after lyophilization overnight.

Glucose, lactate, acetate, butyrate, and ethanol in the broth were monitored with an Agilent 1100 high-performance liquid chromatography (HPLC) system equipped with a refractive index detector (RID) operated at 35 °C and a column (Aminex HPX-87H, 7.8 mm inner diameter and 30 cm length, BioRad Laboratories, Hercules, CA, USA) maintained at 65 °C; 5 mM of $H_2SO_4$ was used as the mobile phase at a flow rate of 0.7 mL/min. Glucose consumption during the whole fermentation process was calculated by subtracting the residual glucose concentration in the broth from the total added glucose. Similarly, the cellulose consumption was also calculated using this method. The mix gas ($CO_2$ and $H_2$) produced in the fermentation was collected in an air-collecting bag and analyzed with a GC system (GC-7820, HuiFen Chemical Instruments Co., Ltd., Zaozhuang, China) equipped with a TDX column (1.5 m length and 3 mm inner diameter). $H_2$ and $CO_2$ were detected using a thermal conductivity detector with high-purity argon as the carrier gas.

## 2.3. Carbon and Energy Flux Analysis

Acids and gas are the main products in *C. cellulovorans* fermentation of cellulose or glucose, and the metabolic pathways have been clarified (Figure 1). Based on the carbon source balance, it is obvious how to calculate the amount of glucose consumed in the formation of acids. However, since accurate measurement of bioenergy is still a challenge, the energy generation and consumption (which mainly refer to ATP, NADH, and $Fd_{red}$ in this study) can only be estimated through glucose consumption and product formation. Bioenergy is mainly produced by glycolysis, during which one mole of glucose can produce two moles of ATP and two moles of NADH in *C. cellulovorans* fermentation. Acetate and butyrate formation can also produce ATP. If the synthesis of acetyl-CoA is catalyzed by pyruvate:ferredoxin oxidoreductase (PFOR), $Fed_{red}$ will be formed in this reaction; if this reaction is catalyzed by pyruvate formate lyase (PFL), there is no $Fd_{red}$ produced. Furthermore, $Fd_{red}$ can be produced during butyrate formation due to the participation of the Bcd/Etf electron bifurcation enzyme complex. In this study, we established that all acetyl-CoA required in the synthesis of acetate and butyrate was catalyzed by PFOR.

Therefore, the molar concentration of $Fd_{red}$ could be calculated through molar concentration of acetate and butyrate.

These forms of bioenergy—mainly ATP and NADH—promote the cell growth and metabolism. Excess energy is reoxidized through product formation to maintain the redox balance during cell metabolism. Lactate synthesis requires NADH (Figure 1). The synthesis of 1 mole of butyrate requires 3 moles of NADH, and 1 mole of $H_2$ consumes 1 mole of $Fd_{red}$ (Figure 1). It should be pointed out that there was still 20 mM of acetate in the original fermentation medium, which should be taken into account when calculating the carbon source balance.

### 2.4. Gene Expression Analysis by RNA-Seq and qRT-PCR

Comparative transcriptomics of cells grown on glucose and cellulose was performed to investigate gene expression profiles based on three biological triplicates. Cell pellets at the late-exponential phase from cultures in the bioreactor were collected via centrifugation at $10,000 \times g$ and $-4\ °C$ for 10 min, frozen in liquid nitrogen immediately, and then stored at $-80\ °C$. The RNA isolation and high-throughput RNA sequencing (RNA-Seq) were accomplished by Allwegenetech Corp. (Beijing, China). The total RNA was extracted using the mirVana miRNA Isolation Kit (Ambion, Santa Clara, CA, USA) following the manufacturer's protocol. RNA integrity was evaluated using the Agilent 2100 Bio-analyzer (Agilent Technologies, Santa Clara, CA, USA). The samples with an RNA Integrity Number (RIN) $\geq 7$ were subjected to subsequent analysis. The libraries were constructed using the TruSeq Stranded mRNA LTSample Prep Kit (Illumina, San Diego, CA, USA) according to the manufacturer's instructions. Then, these libraries were sequenced on the Illumina sequencing platform (HiSeqTM 2500), and 150 bp/125 bp paired-end reads were generated. Based on the reads per kilobase of transcript per million reads mapped (RPKM) normalization, the gene expression profiles were analyzed. The processed RNA-Seq data was submitted to the ArrayExpress database (www.ebi.ac.uk/arrayexpress, URL, accessed on 21 November 2022) under the accession number E-MTAB-12360. The transcriptional expression profiles of some genes were detected via quantitative reverse transcription-PCR (qRT-PCR). Triplicate samples were collected from cultures at the late exponential phase of growth. The collection and storage method of the bacterial pellets and the operation of the qRT-PCR were described in detail in a previous publication [22]. The quantities of transcripts from these genes were normalized with that of 16S rRNA as the internal standard. Relative transcript levels of the studied genes were calculated using the threshold cycle ($2^{-\Delta\Delta CT}$) method [22,23]. The primers used in this study are listed in Table S1 in the Supplementary Materials.

### 2.5. Proteomic Analysis

The extracellular proteome technology was used to compare the metabolism difference of *C. cellulovorans* grown on glucose and cellulose. Extracellular protein samples were harvested via centrifugation ($8000 \times g$, $4\ °C$, 10 min) from glucose- (36 h) and cellulose-supplemented (96 h) cultures, and the supernatant was stored at $-80\ °C$ for the following proteome analysis. Protein digestion and a Sequential Window Acquisition of all Theoretical Fragment Ion Spectra–Mass Spectrometry (SWATH-MS) analysis were conducted by a specialized company (Allwegenetech Corp., Beijing, China) [24]. Briefly, the protein concentration of each sample was determined, and 100 μg protein from each sample was taken out for enzymatic hydrolysis following the detailed description by Yin et al. [25–27].

LC-MS/MS analyses were performed using a Orbitrap Fusion Lumos coupled with an Easy nanoLC 1000 system (Thermo Scientific, Waltham, MA, USA). Peptide (3 μg) was loaded into a home-packed column (3 μm, Waters Bridge C18, 120 Å, 4.6 mm × 250 mm) with an integrated spray tip. The mobile phase was a mixture of 0.1% (*v/v*) formic acid in water (buffer A) and 0.1% (*v/v*) formic acid in acetonitrile (buffer B) at a flow rate of 400 nL/min. The peptides were separated by a 120 min gradient (6% buffer B for 1 min, 36% buffer B for 100 min, 60% buffer B for 106 min, 100% buffer B for 110–118.50 min, then

held at 1% buffer B). To generate SWATH-MS spectral library, the reference spectral library was firstly constructed using the data-dependent acquisition (DDA) mode. Then, a cyclic data-independent analysis (DIA) was also constructed. The software used for the data analysis and setting of the mass spectra parameters was described in detail in previous reports [28,29]. All of the raw data files were deposited in the PRIDE database with the accession number PXD037889.

The mass spectrometry proteomics data provided information about the differential expression of the extracellular protein secreted by *C. cellulovorans* grown on glucose and cellulose. In combination with the whole genome sequence of *C. cellulovorans*, the present study focused on determining: (1) the proteins with the highest expression levels in glucose-supplemented fermentation (the top 60 proteins); (2) the proteins with the highest expression levels in cellulose-supplemented fermentation (the top 60 proteins); and (3) the proteins with the greatest differences in expression levels between glucose- and cellulose-supplemented fermentation [16].

## 3. Results

### 3.1. Growth and Fed-Batch Fermentation

*C. cellulovorans* grew well in the fermentation with glucose as the carbon source: the highest optical density (OD; 600 nm) achieved $3.5 \pm 0.2$ at 48 h. The strain produced $16.8 \pm 5.8$ mM of lactate, $22.3 \pm 7.5$ mM of acetate, and $132.7 \pm 5.5$ mM of butyrate in the present study (Figure 2). A total of $268 \pm 2$ mM of glucose was consumed in the entire fermentation (72 h). $H_2$ and $CO_2$ generated during the fermentation were collected in an air-collection bag. Through calculation, we determined that $117.2 \pm 14$ mM of $H_2$ and $36.8 \pm 3$ mM of $CO_2$ were produced by *C. cellulovorans*, respectively. Furthermore, we drew a standard curve of the OD and total intracellular protein concentration for cells grown in glucose (Figure S1). This standard curve helped us determine the OD of cells in cellulose-supplemented culture. When *C. cellulovorans* grew in cellulose-supplemented culture at pH 7.0, it consumed around 33 g/L cellulose (equivalent to $204 \pm 12$ mM of glucose) and mainly produced 11.5 mM $\pm 1.3$ mM of acetate and $93.9 \pm 4.4$ mM of butyrate by the end of fermentation. Lactate was not detected under this condition. The highest OD achieved $2.6 \pm 0.3$ at 120 h based on determination of the total intracellular cell proteins (Figure 2). The gas productivity was measured as above; the data are shown in Table S2. In comparison, the *C. cellulovorans* grown on cellulose produced fewer acids, the growth rate was much lower than that on glucose, and the fermentation course time was 2.7 times longer (Figure 2).

### 3.2. Carbon Source, Redox Balance Analysis, and Stoichiometry

Regarding the metabolic pathways (Figure 1), acids were predominantly generated by glucose metabolism. Acid formation required 152.3 mM of glucose in *C. cellulovorans* fermentation grown on glucose, while 268 mM of glucose was consumed in the entire fermentation process (Tables 1 , S2 and S3). Therefore, 57% of the carbon source went toward acid synthesis. Similarly, 49% of the total glucose consumed (33 g/L cellulose; equivalent to $204 \pm 12$ mM of glucose) in the *C. cellulovorans* fermentation with cellulose as the carbon source was used to synthesize acids. These results suggested that around half of the carbon source was metabolized to other products in *C. cellulovorans* fermentation. Meanwhile, two methods were used to determine the protein concentration in broth. Around $0.52 \pm 0.1$ g/L and $0.61 \pm 0.2$ g/L of protein were determined using the Bradford method in the supernatant of the broth with glucose and cellulose as carbon sources, respectively; the protein concentrations measured using the lyophilization method were $0.9 \pm 0.1$ g/L and $1.1 \pm 0.2$ g/L. *C. cellulovorans* is a cellulose-degrading bacterium that can secrete extracellular enzymes to obtain soluble sugar in cellulose-supplemented fermentation. However, the results showed that *C. cellulovorans* also produced a large amount of protein in the glucose-supplemented culture in this study.

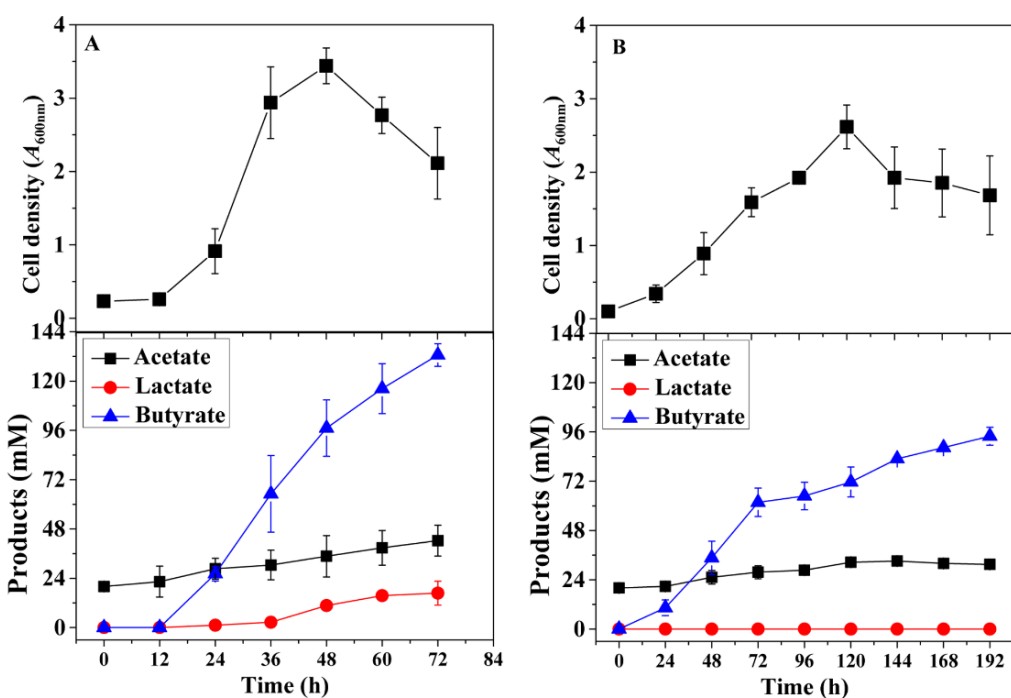

**Figure 2.** Growth and products profiles of *C. cellulovorans* grown on glucose (**A**) and cellulose (**B**).

**Table 1.** Detailed data from the carbon source, redox balance analysis, and stoichiometry in *C. cellulovorans* fermentation.

| | Glucose [a] (mM) | Glucose [b] (mM) | Ratio [c] (%) | RE [d] (mM) | RE [e] (mM) | Ratio [f] (%) |
|---|---|---|---|---|---|---|
| Glucose | 268 ± 2 | 152.3 | 57 | 1072 | 399.4 | 37 |
| Cellulose | 204 ± 8 | 99.7 | 49 | 816 | 329.2 | 40 |

[a] Glucose consumption in the entire fermentation process, during which the molecular weight of glucose in cellulose was 162; [b] glucose required for the acetate, lactate, and butyrate synthesis; [c] ratio of glucose required for acid formation to total glucose consumption; [d] total reducing equivalent (NADH and $Fd_{red}$) formed during glucose metabolism to acetyl-CoA; [e] total reducing equivalent (NADH and $Fd_{red}$) required for acid and hydrogen synthesis during the entire fermentation process; [f] ratio of reducing equivalent to reducing equivalent (e/d).

The synthesis and consumption of bioenergy promote cell metabolism and maintain redox balance in *C. cellulovorans* fermentation. According to the metabolic mechanism that was previously clarified, ATP and NADH were mainly produced in glycolysis (Figure 1). The synthesis of acetyl-CoA and butyryl-CoA could produce $Fd_{red}$. Butyrate and lactate formation consumed NADH, and the hydrogen generation consumed $Fd_{red}$ (Figure 1). Theoretically, around 1072 mM of reducing equivalents (the total NADH and $Fd_{red}$) were produced by the glycolysis pathway in glucose fermentation. The syntheses of butyrate, hydrogen, and lactate required 399.4 mM of reducing equivalents, accounting for 37% of the total reducing equivalents. In addition, the ratio was 40% in the cellulose fermentation (Table 1). Around 536 mM of NADH was produced with glucose as the carbon source in the present study, and the synthesis of butyrate and lactate required 414.9 mM of NADH. Therefore, the NADH required for acid synthesis covered 77% of the total NADH. Similarly, the NADH required for butyrate synthesis accounted for 69% of the total NADH generated in the glycolysis of the cellulose fermentation (Table S3). $Fd_{red}$ was generated along with acetyl-CoA generation, which was the precursor of butyrate and acetate. As a result, it could be calculated that 420.4 mM of $H_2$ was supposed to be generated in the glucose fermentation. However, the hydrogen yield collected in the practical fermentation was 117.2 mM, which meant only 28% of the total $Fd_{red}$ was used for $H_2$ formation in the glucose

fermentation. Correspondingly, 48% of $Fd_{red}$ was used to generate $H_2$ in the condition of cellulose-supplemented culture (Table S3).

These results showed that in the fermentation of the two types of carbon sources, *C. cellulovorans* modified the distribution mechanism of the carbon flux and energy flux to maintain the carbon balance and redox balance, leading to a wide variation in product yields.

### 3.3. Transcriptional Analyses of C. cellulovorans Grown on Glucose and Cellulose

RNA-Seq technology was used to identify key genes specifically associated with glucose or cellulose metabolism in *C. cellulovorans* fermentation. *C. cellulovorans* grown on glucose produced more acids and consumed more carbon sources than that on cellulose (Figure 1 and Table 1). Therefore, the genes involved in glycolysis and acid formation were firstly analyzed; the transcription profiles of these genes are exhibited in Figure 3 and Tables S4 and S5. For the large majority of these genes, the expression levels were higher in glucose fermentation than those in cellulose fermentation. This observation was basically consistent with the previous comparative proteomic results [2]. Then, the transcription profiles of cellulose genes, which are essential for cellulose degrading, were analyzed (Figure 4 and Table S5). The results showed that the expressions of cellulose-degrading genes were largely upregulated in cellulose fermentation compared to that in glucose fermentation.

Furthermore, particular attention was paid to the genes with transcriptional reads per kilobase of transcript per million mapped reads (RPKM) greater than 30 and fold-changes greater than 3 (log2 value greater than 1.6 or less than −1.6). About 103 genes upregulated in cellulose fermentation were selected; these are listed in Supplementary Table S4. Among these, there were 29 cellulose-degrading genes reported in a previous study; these are marked in blue in Table S4. An operon (Clocel_2816-2824) that was predicted to code for cellulose degrading was highly induced in cellulose fermentation. Its RPKM values were very high; some of them were greater than 10,000, indicating that this operon plays an important role in microcrystalline degradation (Table S4). On the contrary, the RPKM values of genes belonging to another operon (Clocel_2593-2600) that was also highly induced in cellulose fermentation were very low, indicating that this operon is only related to cellulose utilization. Based on the gene sequence information, we speculated that this operon was involved in the uptake of oligosaccharides that derived from cellulose depolymerization.

The 79 genes upregulated in glucose were selected; these are listed in Table S4. All of the 79 protein sequences were submitted to a website for predicting signal peptides (http://www.cbs.dtu.dk/services/SignalP-4.1). We found that the proteins encoded by four genes (Clocel_0521, Clocel_4152, Clocel_1183, and Clocel_2675) had signal peptides; these are marked in red in Table S4. Since there was a large amount of protein (0.9 g/L) found in the glucose-supplemented culture, the five proteins may have been the main components of extracellular proteins. However, further verification using a quantitative proteome is required.

About 11 genes were randomly selected from the transcriptome for validation. The expression profiles of these 11 genes were investigated using RT-PCR to verify the RNA-Seq results. The gene information and primers are listed in Table S1, and the RT-PCR results are shown in Figure S2. The results showed that the trends in the change in transcription levels obtained by the two methods were consistent (Figures 3, 4 and S2 and Table S4). The genes used in the RT-PCR are underlined in Table S4.

### 3.4. Quantitative Proteome Analyses of Culture Supernatants in C. cellulovorans

Generally, it is unnecessary for *C. cellulovorans* to produce cellulase in the fermentation with glucose as the carbon source. However, a large amount of protein was detected in the supernatant (around 0.9 g/L). Quantitative proteome analyses of the culture supernatant in glucose and cellulose fermentation were performed to identify the extracellular proteins, and the protein abundance was indicated by spectral counts of the identified peptides.

Therefore, we firstly focused on proteins whose spectral counts were greater than $10 \times 10^{10}$. Then, the protein sequences of these selected proteins were submitted to a website (http://www.cbs.dtu.dk/services/SignalP-4.1) to predict the signal peptides [30]. With the above cutoff criteria, 44 proteins were identified in glucose fermentation (including 3 proteins with signal peptides), and 60 proteins including 14 proteins with signal peptides were identified in cellulose. These proteins are listed in Figure 5 (top 30 in glucose fermentation), Figure 6 (top 30 in cellulose fermentation), and Table S6. Among these proteins, 20 were identical (marked in red in Table S6), indicating that they were expressed in a high abundance under the two fermentation conditions.

| Gene Name | Log2FC | Product | |
|---|---|---|---|
| Clocel_2901 | | ATP-dependent 6-phosphofructokinases | |
| Clocel_0388 | | ATP-dependent 6-phosphofructokinases | |
| Clocel_1603 | | Pyrophosphate (PPi)-fructose 6-phosphate 1-phosphotransferase | |
| Clocel_0719 | | Glyceraldehyde-3-phosphate dehydrogenase | |
| Clocel_0720 | | Phosphoglycerate kinase | Glycolyusis |
| Clocel_0721 | | Triosephosphate isomerase | |
| Clocel_1364 | | Glucose-6-phosphate isomerase | |
| Clocel_1533 | | Lactate dehydrogenase, *ldh* | |
| Clocel_2700 | | Lactate dehydrogenase, *ldh* | |
| Clocel_4097 | | Fe-only hydrogenase, *hyd* | |
| Clocel_1684 | | Pyruvate ferredoxin oxidoreductase, *pfor* | |
| Clocel_1811 | | Pyruvate formate lyase, pfl | |
| Clocel_1812 | | Pyruvate-formate lyase activating enzyme, *pflA E* | |
| Clocel_1891 | | Phosphate acetyltransferase,*pta* | Central metabolism |
| Clocel_1892 | | Acetate kinase, *ack* | |
| Clocel_3058 | | Acetyl-CoA acetyltransferase, *thl* | |
| Clocel_2972 | | 3-hydroxybutyryl-CoA dehydrogenase, *hbd* | |
| Clocel_2973 | | Electron transfer flavoprotein subunit alpha | |
| Clocel_2974 | | Electron transfer flavoprotein subunit beta | |
| Clocel_2975 | | Acyl-CoA dehydrogenase, *bcd* | |
| Clocel_2976 | | Short-chain-enoyl-CoA hydratase, *ech* | |
| Clocel_3674 | | Butyrate kinase, *buk* | |
| Clocel_3675 | | Phosphate butyryltransferase, *ptb* | |
| Clocel_1405 | | Hydratase, aconitase | |
| Clocel_3688 | | Citrate/2-methylcitrate synthase, citrate synthase | TCA circle |
| Clocel_2469 | | Isocitrate/isopropylmalate dehydrogenase family protein | |
| Clocel_0392 | | Class II fumarate hydratase, *fumC* | |
| Clocel_1284 | | NADP-specific glutamate dehydrogenase, *gdhA* | |
| Clocel_2665 | | Glutamate synthase large subunit, *gltB* | |
| Clocel_2992 | | NADPH-dependent glutamate synthase, *gltA* | |

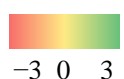

−3  0  3

**Figure 3.** Expression profiles of genes located in the central metabolic pathways (Figure 1). The red color indicates that the gene expression level was higher in glucose-supplemented culture than that in cellulose-supplemented culture. On the contrary, the green color indicates that the gene expression level was higher in cellulose-supplemented culture. The details of the expression profiles are listed in Table S3. Gene name: genes are listed in the order of old ORF (open reading frame) numbers; Log2FC: data represent the $\log_2$ value fold-changes in the reads per kilobase per transcript per million reads mapped (RPKM) during cellulose fermentation as compared to that during glucose fermentation.

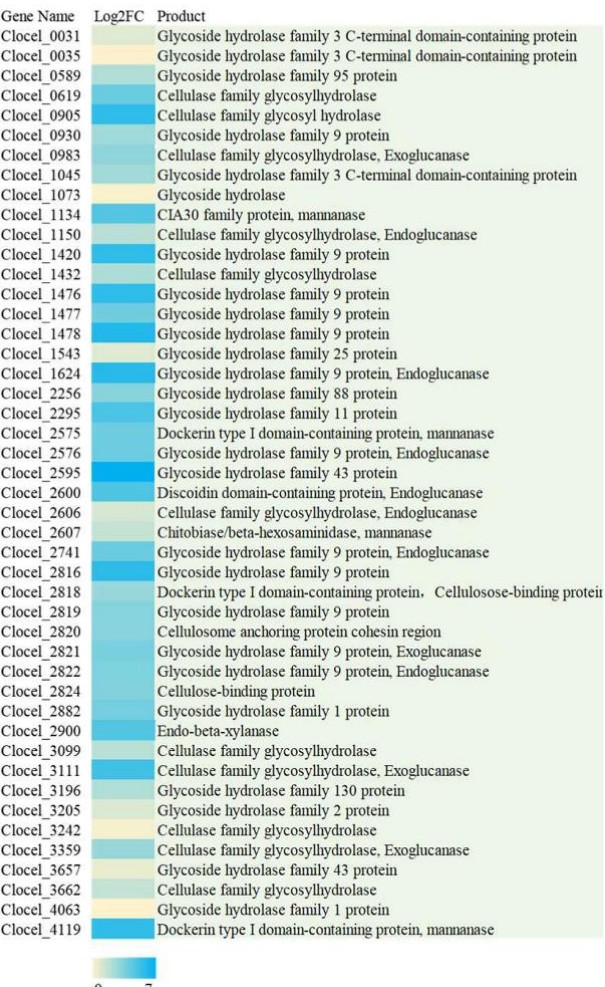

**Figure 4.** Expression profiles of genes involved in cellulose degrading. The blue color indicates that the gene expression level was higher in cellulose-supplemented culture than that in glucose-supplemented culture. The details of the expression profiles are listed in Table S3. Gene name: genes are listed in the order of old ORF (open reading frame) numbers; Log2FC: data represent the $\log_2$ value fold-changes in the reads per kilobase of transcript per million mapped reads (RPKM) during cellulose fermentation as compared to that during glucose fermentation.

*C. cellulovorans* can produce extracellular enzymes to obtain soluble fermentation sugar in cellulose fermentation. Hence, in cellulose fermentation, many proteins with high abundance are involved in polysaccharide depolymerization and sugar transport. We found that 7 out of 60 high-abundance proteins (gene number underlined in Table S6) belonged to this class. In addition, some proteins were discovered to participate in butyrate synthesis (e.g., the products of the genes Clocel_1684, Clocel_2973, and Clocel_2974; Figure 1). Furthermore, some Fe-S-dependent proteins were also identified (Table S6), which suggested that iron and sulfur sources played an important role in metabolism of *C. cellulovorans*.

In the glucose fermentation, only 3 proteins (Clocel_2823, Clocel_2824, and Clocel_2900) of the selected 44 high-abundance proteins had signal peptides. The three proteins were all involved in plant polysaccharide depolymerization, indicating that they were constitutively expressed even if there was no lignocellulose substrate in the medium. Moreover, the analysis data revealed that the proteins involved in butyrate biosynthesis were abundant, including pyruvate:ferredoxin oxidoreductase (PFOR, Clocel_1684, and Clocel_2840), pyruvate formate lyase (PFL and Clocel_1811), and the Bcd/Etf complex (Clocel_2972, Clocel_2973, and Clocel_2974). These enzymes were located in the central metabolic path-

way and upregulated in glucose fermentation (Figure 3 and Table S3). The quantitative proteome data listed in Table S6 also revealed that the majority of proteins were not secreted proteins but intracellular proteins. The possible reason why these proteins were present in the supernatant of the fermentation broth was that they were released with cell autolysis. The data also showed that the proteins involved in butyrate synthesis occupied a large proportion of the total intracellular proteome, especially in the fermentation with glucose as the carbon source (Table 1 and Figure 3).

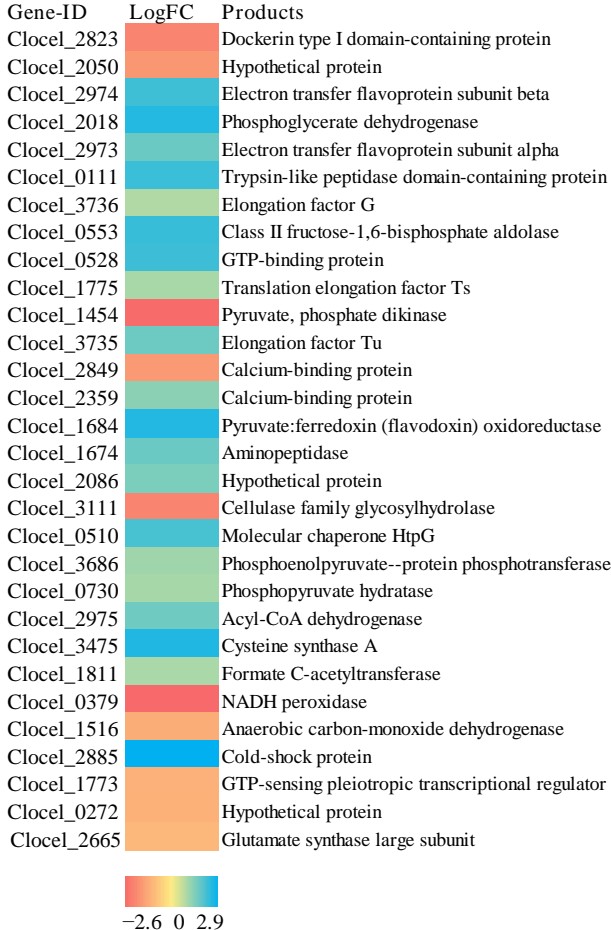

**Figure 5.** Expression profiles of proteins (top 30 in spectral counts) in glucose fermentation. The red color indicates that the protein expression level was higher in cellulose-supplemented culture than that in glucose-supplemented culture. On the contrary, the green color indicates that protein expression level was higher in glucose-supplemented culture. The details of the expression profiles are listed in Table S4. Gene ID: genes are listed in the order of old ORF (open reading frame) numbers; LogFC: data represent the $\log_2$ value fold-changes in the spectral counts during cellulose fermentation as compared to that during glucose fermentation.

The protein expression profiles of the four genes (Clocel_0521, Clocel_4152, Clocel_1183, and Clocel_2675) with signal peptides and a high transcription level are listed in Table S4 (marked in blue). The protein expression level of three genes (Clocel_4152, Clocel_1183, and Clocel_2675) had little change in the fermentation using glucose or cellulose as the carbon source. The gene (Clocel_0521) coding for a sulfate transporter had a higher level of transcription and expression in glucose fermentation than that in cellulose fermentation. However, its RPKM value and spectral counts were not outstanding (Tables S4 and S5), indicating that the proportion of this protein in the total extracellular proteins was not high in glucose fermentation.

| Gene-ID | LogFC | Products |
|---|---|---|
| Clocel_2050 | | Hypothetical protein |
| Clocel_2823 | | Dockerin type I domain-containing protein |
| Clocel_1454 | | Pyruvate, phosphate dikinase |
| Clocel_0379 | | NADH peroxidase |
| Clocel_3111 | | Cellulase family glycosylhydrolase |
| Clocel_2849 | | Calcium-binding protein |
| Clocel_2900 | | Endo-1,4-beta-xylanase |
| Clocel_2596 | | Sugar ABC transporter substrate-binding protein |
| Clocel_1684 | | Pyruvate:ferredoxin oxidoreductase |
| Clocel_1516 | | Carbon-monoxide dehydrogenase |
| Clocel_4153 | | Rubrerythrin family protein |
| Clocel_1773 | | GTP-sensing pleiotropic transcriptional regulator |
| Clocel_0272 | | Hypothetical protein |
| Clocel_4154 | | Desulfoferrodoxin |
| Clocel_2607 | | Chitobiase |
| Clocel_2402 | | Acetaldehyde/alcohol dehydrogenase |
| Clocel_3817 | | Iron-containing alcohol dehydrogenase |
| Clocel_2665 | | Glutamate synthase large subunit |
| Clocel_3196 | | Glycoside hydrolase family 130 protein |
| Clocel_0391 | | N,N'-diacetylchitobiose phosphorylase |
| Clocel_2006 | | Translation initiation factor IF-3 |
| Clocel_0038 | | ABC transporter substrate-binding protein |
| Clocel_3736 | | Elongation factor G |
| Clocel_0148 | | Leucine-rich repeat protein |
| Clocel_0590 | | Xylose isomerase |
| Clocel_2818 | | Dockerin type I domain-containing protein |
| Clocel_4143 | | Acyl carrier protein |
| Clocel_2744 | | RidA family protein |
| Clocel_2741 | | Glycoside hydrolase family 9 protein |
| Clocel_0146 | | Protease inhibitor I42 family protein |

−3.5　0　1.9

**Figure 6.** Expression profiles of proteins (top 30 in spectral counts) in cellulose fermentation. The red color indicates that the protein expression level was higher in cellulose-supplemented culture than that in glucose-supplemented culture. On the contrary, the green and blue colors indicate that the protein expression level was higher in glucose-supplemented culture. The details of the expression profiles are listed in Table S4. Gene ID: genes are listed in the order of old ORF (open reading frame) numbers; LogFC: data represent the $\log_2$ value fold-changes in spectral counts during cellulose fermentation, as compared to that during glucose-fermentation.

## 4. Discussion

### 4.1. Growth and Metabolic Product Characteristics

The growth in *C. cellulovorans* was greatly affected by the substrate in the fermentation. *C. cellulovorans* grew fast and metabolized rapidly in soluble sugar fermentation, but it had a low growth rate and a long fermentation period in lignocellulose and pectin fermentation [9,15,17]. In this study, the maximum OD value was achieved at 48 h in

glucose-supplemented fermentation; while in cellulose-supplemented fermentation, it appeared at 120 h. Similar findings were reported in a previous study [2]. Due to the solid structure and insolubility of natural lignocellulose, it must be digested into soluble sugars by enzymes before the fermentation. All of the enzyme production, secretion, combination with the substrate, and subsequent enzymatic hydrolysis will slow down the acquisition time of soluble sugars, resulting in reduced growth and metabolic rates [3,10]. This phenomenon occurs in the fermentation of all lignocellulose-degrading bacteria [31,32]. These results showed that the bioconversion of lignocellulose to biochemical was still limited by the enzymatic hydrolysis rate of lignocellulose.

The optimal pH range of *C. cellulovorans* fermentation is very narrow, which means that the pH in fermentation plays a very important role in the growth and metabolism [2,9,10]. In this study, the pH value was performed at 7.0, which is also the optimal pH of *C. cellulovorans*. Under this condition, acetate and butyrate were mainly produced. If conducted without pH control, when the pH was reduced to 6.0, *C. cellulovorans* no longer grew or metabolized glucose. Meanwhile, ethanol was synthesized, indicating that *C. cellulovorans* redistributed the carbon source and energy flux in response to the pH change in the external environment. There is a more obvious phenomenon showing that pH affects product synthesis in the ABE-fermentation of *C. acetobutylicum* [12,33]. The main products are acids in the high pH fermentation, which is called acidogenesis; meanwhile, the main products change to alcohols in the low-pH fermentation, which is named solventogenesis. In particular, this conversion mechanism from acidogenesis to solventogenesis has not been completely clarified. Obviously, alcohol formation requires more NAD(P)H than acid formation (Figure 1), which suggests that the production and distribution mechanisms of NAD(P)H should be firstly clarified to understand the relationship between the conversion mechanism from acid formation to alcohol production with the pH value change in the *Clostridium* species fermentation [5].

The addition of key elements to the medium and the application of appropriate fermentation technology can also affect the growth and product yield in *Clostridium* species fermentation. In this study, we used fed-batch fermentation technology and increased the concentration of the sulfur source (Cys-HCl) in the medium. As a result, *C. cellulovorans* consumed 33 g/L cellulose, which was much higher than the previously reported consumption of 17 g/L [9]. The high cellulose consumption greatly increased the butyrate titer, which achieved 8.1 g/L in the cellulose fermentation of *C. cellulovorans*. Similarly, the glucose consumption and butyrate titer in this study were also higher than those previously reported [2]. The high product titer obtained in this study provides an obvious advantage to study the metabolisms in different fermentation conditions of *C. cellulovorans*.

### 4.2. Preliminary Analysis of Carbon Source Distribution Mechanism

Based on the carbon source balance analysis, we found that the protein was also a main product whose synthesis occupied a large amount of carbon sources derived from glucose metabolism in *C. cellulovorans* fermentation. The precursors of acid (butyrate and acetate) synthesis such as pyruvate and acetyl-CoA, which are important intermediates in glucose metabolism, are also precursors in amino acid synthesis (Figure 1). Therefore, the protein synthesis and acid generation compete in the utilization of the carbon source. However, the distribution mechanism of the carbon source is still unclear. Different substrates can also affect the carbon flow in *C. cellulovorans* fermentation. The glucose and nutrient elements are abundant in glucose fermentation of *C. cellulovorans*. Hence, the strain accumulates biomass rapidly and requires a great deal of ATP for amino acid and nucleic acid synthesis. Butyrate and acetate formation can provide ATP in addition to glycolysis (Figure 1). As a result, the carbon source is mainly distributed for acid formation in glucose fermentation of *C. cellulovorans*. In this study, the glucose for acid formation accounted for 67% of the total glucose consumption; meanwhile, this ratio reached 98% in glucose fermentation in previous reports [2]. On the contrary, the carbon source was preferentially used for cellulase synthesis to obtain soluble sugars for growth and metabolism in the cellulose

fermentation of *C. cellulovorans*. Therefore, the acid titer and ATP synthesis rate decreased, resulting in a lower growth rate compared to glucose fermentation. The comparison of the transcriptome and proteome results further verified the fermentation results at the transcription and translation levels, which indicated that different substrates can affect the carbon source distribution in the fermentation of *C. cellulovorans*.

Interestingly, a large amount of protein was also determined in the supernatant in the glucose fermentation of *C. cellulovorans.* The majority of these proteins had no signal peptide; some proteins such as PFOR, PFL, and the Bcd/Etf complex were involved in butyrate synthesis (Figure 3 and Table S4). We speculated that these proteins were released though cell autolysis of *C. cellulovorans*, indicating that the butyrate fermentation process was accompanied by cell autolysis (Figure 2). Cell autolysis is a significant characteristic in bacteria fermentation; this phenomenon is caused by nutrient deficiency or fermentation environmental changes [34–37]. In this study, the fermentation temperature and pH remained stable. Thus, we speculated that the deficiencies in some certain nutrient elements caused autolysis of *C. cellulovorans*.

### 4.3. Preliminary Analysis of Redox Balance Mechanism

NADH and $Fd_{red}$ are the main reducing equivalents produced during glucose metabolism in *Clostridium* species fermentation [12,38]. They cannot be accumulated in cells and must be oxidized immediately to maintain the redox balance in cells. Butyrate synthesis can oxidize NADH to generate $NAD^+$, and $H_2$ formation is a main pathway for $Fd_{red}$ oxidization (Figure 1). The data showed that the NADH required for butyrate formation accounted for 77% of the total NADH produced by glycolysis in glucose fermentation, which was higher than that (69%) in cellulose fermentation (Table 1). Moreover, the ratio of $Fd_{red}$ required for $H_2$ formation (48%) to total $Fd_{red}$ produced in acid synthesis was higher in cellulose fermentation than that (28%) in glucose fermentation (Table S3). These results suggested that $Fd_{red}$ participated in metabolic activities other than $H_2$ formation in *C. cellulovorans* fermentation. Based on our knowledge, the specific metabolic reaction in which $Fd_{red}$ participates is still unclear [12]. In the anaerobic bacteria and archaea, many ferredoxin:$NAD^+$/$NADP^+$ oxidoreductase (FNOR) enzymes or complexes have been found and recognized as a key bridge in NADH/NADPH formation via electron bifurcation or direct electron transfer from $Fd_{red}$ [13,39]. Those FNOR enzymes with specific functions (including the Nfn complex and Rnf complex) were not found in *C. cellulovorans*. Recently, a ferredoxin:$NAD^+$ reductase (CA_C0764) and a ferredoxin: $NADP^+$ reductase (CA_C1502) were confirmed in *C. acetobutylicum*, and similar enzymes were found in the genome of *C. cellulovorans* according to transcriptome data [13]. These are encoded by the genes of Clocel_1284 and Clocel_2665, whose transcription levels were high in both glucose and cellulose fermentation (Figures 3 and 4). Another NFOR enzyme confirmed in *Thermoanaerobacterium saccharolyticum* (Tsac_1705) was also found in *C. cellulovorans* (Clocel_1556) [39]. This is a dihydroorotate dehydrogenase electron transfer subunit and is a part of a putative operon for de novo synthesis of pyrimidine. The redox balance analysis data indicated that $Fd_{red}$ indirectly participated in cell metabolism by producing NAD(P)H, which is necessary for the synthesis of amino acids and nucleic acids in microbial growth.

In this study, it was assumed that the acetyl-CoAs required for acid formation were all catalyzed by PFOR from pyruvate. However, acetyl-CoA can be synthesized by PFL catalysis of pyruvate (Figure 1). The omics results showed that the transcriptional and expression levels of PFL and PFOR were very high in both glucose and cellulose fermentation (Figure 3 and Table 1). It is still a challenge to determine the percentage of acetyl-CoA that is catalyzed by PFOR using existing methods. As a result, it was difficult to accurately calculate how much $Fd_{red}$ was generated at this step in the *C. cellulovorans* fermentation. Nevertheless, this had little impact on the analysis and comparison of the energy balance under different carbon sources. In this study, the results obviously indicated that the $Fd_{red}$ required for cellulase synthesis was less than that required for the formation of key enzymes located in the central metabolic pathways (Tables 1, S2 and S3).

## 5. Conclusions

The growth and product yield in *C. cellulovorans* fermentation are highly dynamic. Here, two representative carbon sources (glucose and cellulose) were used to explore the metabolic regulation mechanism. In the cellulose fermentation, *C. cellulovorans* secreted carbohydrate activity enzymes to obtain soluble sugars. The enzymatic hydrolysis rate restricted bioconversion from lignocellulose to biochemical, resulting in a low growth rate and a long fermentation period. The protein was also found to be a main product whose synthesis consumed a large amount of carbon sources and energy sources. Even in glucose fermentation, a large amount of protein was detected in the supernatant. Most of these proteins participated in butyrate synthesis and did not have signal peptides, indicating that the butyrate generation was accompanied by cell autolysis. Both protein and acid syntheses consumed reducing powers to maintain the carbon source balance and redox balance. Overall, protein was a significant product of *C. cellulovorans* fermentation, which provides a useful framework for further study on metabolic regulation and high-yield biochemical production.

**Supplementary Materials:** The following supporting information can be downloaded at: https://www.mdpi.com/article/10.3390/fermentation9040321/s1, Figure S1: The standard curve of the corresponding relationship between absorbance (A600) and intracellular protein concentration; Figure S2: Expression profiles of selected genes using RT-PCR. Gene name: Genes are listed in the order of old ORF (open reading frame) numbers; Table S1: The primers used for RT-PCR in this study; Table S2: The detailed products data in the end of fermentation of *C. cellulovorans*; Table S3: The detailed data of carbon source, redox balance analysis and stoichiometry in *C. cellulovorans* fermentation; Table S4: The expression profiles of the genes strongly induced during cellulose- and glucose-fermentation of *C. cellulovorans*; Table S5: The expression profiles of the important genes during cellulose- and glucose-fermentation of *C. cellulovorans*. Table S6: The expression profiles of the protein induced during cellulose- and glucose-fermentation of *C. cellulovorans*.

**Author Contributions:** Conceptualization, Z.-Y.L. and F.-L.L.; methodology, W.-Z.T.; software, Y.-X.F.; validation, Q.Z., D.-D.J. and L.-C.L.; formal analysis, D.-D.J.; investigation, Z.-Y.L.; resources, F.-L.L.; data curation, Y.-X.F.; writing—original draft preparation, Z.-Y.L.; writing—review and editing, W.-Z.T.; visualization, L.-C.L.; supervision, Q.Z.; project administration, Z.-Y.L.; funding acquisition, F.-L.L. All authors have read and agreed to the published version of the manuscript.

**Funding:** This work was financially supported by the National Natural Science Foundation of China (Grant No. U21B2099), the Basic Research Project of Colleges and Universities of the Educational Department of Liaoning Province (Grant No. LJKMZ20220894), and the China Petrochemical Corporation (Sinopec).

**Institutional Review Board Statement:** Not applicaple.

**Informed Consent Statement:** Not applicaple.

**Data Availability Statement:** The RNA-seq data had been submitted to EMBL-EBL with Accession No. E-MTAB-12360. All of the raw data files of Mass Spectrometry were deposited in the PRIDE database with the Accession number PXD037889.

**Conflicts of Interest:** The authors declare that they have no known competing financial interest or personal relationship that could have appeared to influence the work reported in this paper.

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
