# Peer review of "Metabolic Difference Analysis of Clostridium cellulovorans Grown on Glucose and Cellulose"

_fermentation, doi:10.3390/fermentation9040321_

Round 1

Reviewer 1 Report

The study by Tang and co-workers reports the results of a comparative analysis of the metabolism of C. cellulovorans growing on glucose or crystalline cellulose. Although similar studies have previously been performed, the present investigation takes advantage from the utilization of pH controlled cultures, which better uncouples the effect of different carbon sources from that determined by pH changes.

In general, the study is sound, interesting and clearly described. However, there is some inaccuracy, especially as concerns calculations and speculation about carbon and energy fluxes, as detailed below.

Major points

1.      Page 2, lines 8-10. Please introduce a reference here.

2.      Page 3, line 5-7. Actually, some non-cellulosomal enzymes have cellulolytic activity (e.g. EngD) or cellodestrin hydrolytic activity (e.g. BglA)

3.      Section 2.3. Carbon and energy flux analysis. The likely existence of FNOR reactions (that is ferredoxin-NAD oxidoreductases) as in other clostridia should be mentioned so that NADH and Fd red pools are not completely separated from each other.

4.      2.4. Gene expression analysis by RNA-Seq and qRT-PCR, line 20-21. Which publications ?

5.      3.1. Growth and fed-batch fermentation, line 3. 22.3 mM acetate ?

6.      Section 3.2. Carbon source, redox balance analysis and stoichiometry. As previously mentioned in my comment on section 2.3, NADH and Fed red pools cannot be considered as separated. I suggest authors to calculate the total amount of reducing equivalents generated by acid production instead and consistently theroretical production of H2. This would add clarity to calculations.

7.      Fig. 5. It is not clear if positive fold changes are referred to genes overexpressed in cellulose- or glucose-grown cultures

8.      Section 3.4, line 11-12. Fig. 5 and 6 should be fig. 6 and 7. The contribution of the results of this section to the overall significance of this study is doubtful. The comparison of the extracellular proteome of C. cellulovorans cultures grown on glucose or polysaccharides was the object of several previous studies. Some of the most significant results of the present investigation are actually referred to intracellular proteins which were likely release by cell autolysis. I suggested this section could be significantly shortened.

9.      Section 5 (also also throughout the manuscript). Actually, the presence of proteins lacking a signal peptide in the extracellular medium does not necessarily means that they are a contamination by intracellular proteins. In several secreted proteins (also including some cellulases)  a signal peptide was not detected.

Minor points

1.      Abstract line 8. Cellulose fermentation ?

2.      Abstract Line 9. 57% of the glucose was consumed to form acids

3.      Page 3, line 4. Expressed not expressing

4.      Section 2.1. The culture medium and batch fermentation, line 24. Anaerobic environment

5.      Section 4.1, line 25. It seems there is an “And” which should not be there

Reviewer 2 Report

The authors performed fed batch cultures of Clostridium cellulovorans, which anaerobically produces butyrate, using glucose or cellulose as the carbon source, and investigated the mass balance and redox cofactor balance, and also measured the transcriptome and proteome. They found that the cells secrete proteins not only in the cellulose fermentation but also in the glucose fermentation, and assumed that this is due to cell autolysis. I think the advantage of using this strain is that it can directly utilize cellulose, so I am not sure why they are focusing on secreted proteins in the glucose fermentation. The purpose of this study should be more clarified. Also, no evidence is provided for the cell autolysis is occurring in the glucose fermentation. My comments are listed below.

Page 1, Line 14 in abstract: "then" is overlapped twice.

Page 6, Line 21: 22.3 ± 7.5 mM of what?

Page 6, Line 22: Time course of glucose consumption should be shown in Fig 2.

Fig. 2: Why is the acetate present from the start of the fermentation?

Page 7, Line 11: Why are the results different between the Bradford and Lyophilization methods?

Table 1: As the authors discuss, the carbon recoveries are not high. Since the conditions are anaerobic, it should be possible to track where the carbon goes. Is the amount used for biomass significant? Are there any metabolites that are not measurable?

Page 11, Line 1: quantitative proteome? Is it typo of RT-PCR analysis?

Fig. 3: The gene expression levels are more informative if they are shown on metabolic pathway map.

Fig. 5 should be aligned with the RNA-Seq results for each gene for easy comparison.

Page 14, Line 5: The 36 h in a glucose carbon source is still in the growth phase. Evidence of cell autolysis should be provided.

The detected extracellular proteins are the types of proteins that are abundant in the cells?

Why did the cell autolysis only occur in the glucose carbon source?

Were there any changes in gene expression related to cell autolysis in the transcriptome data?

Page 15, Line 3: Is this butylate productivity from cellulose superior to other species?
